# An Integrated Indicator to Analyze Sustainability in Specialized Dairy Farms in Antioquia—Colombia

**Gloria P. Rios** [1,*] and **Sergio Botero** [2,*]

[1] Departmento de Ciencias Forestales, Facultad de Ciencias Agrarias, Universidad Nacional de Colombia—Sede Medellín, Medellín, Antioquia 050034, Colombia
[2] Departmento de Ingeniería de la Organización, Facultad de Minas, Universidad Nacional de Colombia—Sede Medellín, Medellín, Antioquia 050034, Colombia
[*] Correspondence: gprios@unal.edu.co (G.P.R.); sbotero@unal.edu.co (S.B.)

**Abstract:** An integrated sustainability indicator in specialized dairy index (IISLE) is proposed for the evaluation of these production systems in four dimensions (financial, technical, social, and environmental). This index was validated in a study carried out with information from 18 dairy farms in Antioquia, a Colombian milk production region. The analytic hierarchy process (AHP) method was applied to prioritize criteria and sub-criteria. According to the proposed dimensions, four criteria were established (financial (ISDF), technical (ISDT), social (ISDS), and environmental (ISDA)) and 30 sub-criteria were analyzed. A multiple correlation analysis was performed to determine the intensity of the relationship among these four criteria and the overall index. The methodology includes a cluster analysis for the calculated IISLE. R software was used in the analysis. The proposed indicator is a powerful and useful planning and control tool for the evaluation of sustainability levels in dairy farms and is highly influenced by financial and technical criteria.

**Keywords:** sustainability indicators; multi-criteria decision-making; specialized dairy

## 1. Introduction

From the standpoint of sustainability, specialized dairy production systems (SDPSs) need to guarantee adequate social–economic conditions for family groups, to persist over time, to obtain good-quality products, and to operate in an environmentally friendly manner in order to maintain or even improve the base of existing natural resources within the system [1–7]. This means that the approach used to quantify sustainable development must be multi-dimensional in nature and include technical, financial, environmental, and social criteria.

In this way, governments need to develop support policies for dairy producers. These policies must be coherent, designed to achieve multiple objectives, effective, and efficient in order to foster competitiveness and sustainability [5,8–12].

When analyzing the sustainability of livestock systems, it is essential to incorporate several criteria [6,13–15]. Producers in this type of business constantly seek to be more competitive and sustainable, increase their economic, financial, technical, environmental, and social efficiency, as well as improve their decision-making capabilities [16–20].

The theory of multi-criteria decision-making has been extensively studied in science, economics, and engineering. The multi-criteria decision analysis (MCDA) is an integrated sustainability assessment methodology suitable for addressing complex problems with a high degree of uncertainty, conflicting objectives, and multiple interests and perspectives [20]. In environmental protection, MCDA has become a tool of great help to production systems and it has also been applied in other systems of a social, economic, agricultural, industrial, ecological, or biological nature [21,22]. The analytic

hierarchy process (AHP) method has also been widely used. The AHP method has a broad range of applications, including in the allocation of human resources, the economy, transport administration, and sustainability in energy systems, agriculture, and industry and can be used in combination with other methods. The AHP method is particularly useful when a numerical value (utility, score) needs to be assigned to each potential action or alternative [23].

Herein, we propose an integrated sustainability indicator in specialized dairy (IISLE, from its Spanish name "Indice Integrado de Sostenibilidad en Lechería Especializada"), which includes relevant criteria identified in the literature. The remainder of this article is organized as follows. Section 2 describes the adopted multi-criteria decision-making approach (analytic hierarchy process, AHP), then presents the development of the criteria and sub-criteria used to construct the index. Section 3 describes the method's application in a case study of dairy farms in a Colombian dairy production region (18 farms in Antioquia) and the data collection process. Section 4 presents an analysis of the results, which includes a cluster analysis of the alternatives and a correlation analysis of the developed criteria index.

## 2. Materials and Methods

This section describes the methodology used to develop the IISLE index. It includes the selected MCDA method (AHP), the criteria and sub-criteria selected for the index, and the normalization method used to grade the criteria and calculate the overall index.

### 2.1. Analytic Hierarchy Process

The AHP method is a tool that allows decision-making to be guided by valuations assigned to a studied phenomenon according to a choice rule [7,24]. These evaluations are carried out by means of discrete and continuous paired comparisons between criteria and sub-criteria, where a real measure can be established or a fundamental scale can be created that reflects the relative strength of preferences and feelings [24]. Its representation starts from factors that, once selected, are organized in a hierarchical structure, which descends from a general objective to criteria, sub-criteria, and alternatives at successive levels, in such a way that captures the relative importance of each of the alternatives in the general objective of the hierarchy [24–27]. The hierarchical analytical process also helps avoid over-simplification and identify and evaluate costs and benefits [7].

AHP breaks down the analysis into criteria and sub-criteria, which are calculated with a weighted average of the scores assigned to each analyzed alternative. In order to obtain the weights for each criterion and sub-criterion based on allocating weights to each of the criteria, Saaty proposes a pairwise comparison in a structured matrix system to obtain an overall rank of weights, in which consistency (defining the relative importance of one criteria with respect to another, and comparing the relative importance to that of other criteria) can be measured [24,25].

### 2.2. Criteria and Sub-Criteria

Four main criteria (environmental, social, financial, and technical) were used to build the IISLE. These criteria were selected after reviewing studies [1,5,9,11,12,19,28–35] related to critical aspects of specialized dairy production systems and consulting professionals in the field of research. The information used to obtain the scores to be assigned to the criteria and sub-criteria came from field surveys and specific production data. Table 1 shows the selected criteria and sub-criteria along with the method used to obtain information and the variables related to each item.

For the financial criteria, the livestock in production define the level of productivity per cow, production costs, and animal loads [36]. The main characteristic of these production systems is the relatively intensive use of production resources which is required to obtain a high level of biological productivity per animal unit or area unit, measured in liters of animal milk/day, although high production costs also imply reduced profit margins per unit produced [18,19,32,37]. On a financial level, livestock farms generate income that is intended to be maximized per cow in production.

Moreover, it was assumed that there would be a secure market for the sale of the products. However, it is important to mention that much capital and high production costs are involved in supplementation, fertilization and pasture management, technical assistance, and permanent labor, etc. [38].

**Table 1.** Criteria, sub-criteria, determination method, and data collection method.

| Criteria and Sub-Criteria | Acronym | Method |
|---|---|---|
| **Financial** | | |
| Average production (PP) cow/year, determined from farm records. | ScF1 | Semi-structured survey. |
| Cost per liter of milk, production cost cow/year (CTP/cow), and average cow-year production (PP) were determined. | ScF2 | CTP-cow-year/PP-cow-year. |
| B/C ratio, determined using gross income. | ScF3 | Gross income-cow-year/CTP-cow-year. |
| Gross margin (MB), determined by cow/year. | ScF4 | Gross income/cow/year—CTP/cow/year. |
| Profitability, determined by cow/year. | ScF5 | MB/productive assets (AP)—cow. |
| Balance point (BP), determined using production cost cow/year and sale price per liter of milk (pvu). | ScF6 | CTP-cow-year/pvu. |
| Production cost cow/year, registration of cost items on farms. | ScF7 | Costing method absorption. |
| **Technical** | | |
| Liters cow/day, cow production farm records. | ScT1 | Semi-structured survey. |
| Interval between births, technical records of farms. | ScT2 | |
| Open days, technical records of farms. | ScT3 | |
| Farm/year deliveries, technical records of farms. | ScT4 | |
| Tuberculosis and brucellosis certificates, physical certificate identification. | ScT5 | |
| Good livestock practices, physical certification. | ScT6 | |
| Milk/concentrate ratio, the consumption of concentrate was identified in the records according to animals in production and the PP. | ScT7 | |
| **Social** | | |
| Quality of life, perception scale between 1 and 12. | ScS1 | Semi-structured survey. |
| Generational relay, directly from the producer. | ScS2 | |
| Social consensus, perception scale between 0 and 9. | ScS3 | |
| Market access, perception scale between 1 and 6. | ScS4 | |
| Institutional support, perception scale between 1 and 12. | ScS5 | |
| Access to credit for facilities, perception scale between 1 and 8. | ScS6 | |
| **Environmental** | | |
| Protection and conservation of water sources, direct observation at sites of streams and wetlands on analyzed farms, ordinal measurement, where a numerical value of 0–4 is assigned to different degrees of the presence of a protective forest. | ScA1 | Likert scale, validated in the field. |
| Occupation period, average occupation period, consulting the monitoring records in the database for the different pastures of each farm. | ScA2 | Semi-structured survey. |
| Rest period, average rest period, consulting the monitoring records in the database for the different pastures of each farm. | ScA3 | Semi-structured survey. |
| Load capacity was obtained by dividing the total area of grassland by the average number of animals in production for each herd according to the data recorded by the farm. | ScA4 | Semi-structured survey. |

**Table 1.** *Cont.*

| Criteria and Sub-Criteria | Acronym | Method |
|---|---|---|
| Erosion, five random pastures were selected at each farm, within which the rest period began. In each pasture, two diagonal routes were made, and five observation points were taken for a total of 25 observations per farm. The number of observations per characteristic of the erosion process and the weighted average of qualification were found for each farm. | ScA5 | Likert scale, validated in the field. |
| Pesticide use, the number of applications, and the quantities applied were corroborated by reviewing the purchase records during the study period and the pesticide inventory at the date of data collection. | ScA6 | Semi-structured survey. |
| Soil conservation practices, direct observation in the field. | ScA7 | Ordinal scale between 0 and 2. |
| Excreta treatment, direct observation in the field. | ScA8 | Ordinal scale between 1 and 3. |
| Destination of wastewater, direct observation in the field. | ScA9 | Ordinal scale between 1 and 3. |
| Greenhouse gas emissions, obtained from the emission factor of each breed present in the herd. | ScA10 | kgCO2eq/kgECM emissions. |
| The density of worms was measured by digging in the ground a hole with dimensions of 15 * 15 * 15. The total number of worms in the volume of extracted earth was obtained. The data are expressed as number of worms/m$^2$. | ScA11 | Random sampling in a zigzag, with 10 repetitions at each farm. |

With respect to the environment, although livestock production is a growing activity in the area it occupies, it creates negative effects. These include deforestation, contamination of water sources, damage to soil, a reduction in biological diversity, global warming, and depletion of the ozone layer, etc. [39–41]. The main environmental impact generated by specialized dairy farms is soil compaction resulting from the transit of animals and causing the erosion of terraces, landslides, or avalanches [37,42,43]. This problem can be managed by controlling the periods during which paddocks are occupied and at rest and also by controlling their carrying capacity. The loss of forest cover is associated with changes in water processes and erosion.

The excreta treatment was assessed with direct observations in the field, including odor issues. The contamination of water sources is caused by excreta, wastewater, and the use of pesticides, etc. [29]. In these production systems, it is common to use animal excreta, which are important resources for recycling and replenishing soil nutrients and carbon, and incorporated into pastures in order to mitigate possible impacts on the environment [44]. All of these types of degradation have led to an accelerated and irreversible loss of soil that has curtailed productivity and led to more expensive, less competitive, and unsustainable livestock over time [5,12,32,45]. The protection and conservation of water sources and soil may mitigate these degradation processes. With respect to greenhouse gas (GHG) emissions, differences in carbon footprints among cattle have been reported in the literature (e.g., 0.8 kgCO2eq/kg ECM for jersey herds and 0.96 kgCO2eq/kg ECM for Holstein herds) [46]. Although cattle are relevant GHG emitters, in the specific case of Colombia (a country with relatively low GHG emissions) the weight given in this study by the experts to this sub-criterion was relatively low in comparison with that given to the other sub-criteria, and it had little effect on the final ranking.

Several issues are relevant to the social criteria. The first is the quality of life of people related to a dairy farm (owners and workers). These systems are structured to be run by owners, who in many cases are accompanied by a butler and some operators. According to [32,47], the following are relevant variables: The educational level of the owners, the technological level, and workers' stability and access to social services [8,31,45,48]. These variables were considered in the survey in order to obtain the respondents' perception of their quality of life. On the other hand, generational relay is a key factor in the sustainability of these systems. In small- and medium-size producers, it is necessary to encourage it in order to take advantage of older producers' experience and knowledge and combine them with younger producers' strength and new ideas [30,38]. Associations among producers

allow them to improve relations and create a social consensus. Other relevant sub-criteria are market access (mainly guaranteeing that their product will be bought), institutional support (related to the governmental and non-governmental institutions in place), and access to credit from financial entities in order to maintain and increase operations [5,32].

Regarding technical criteria, several sub-criteria must be analyzed. First, it is important to determine how efficiently herd reproduction is managed in livestock, since this largely determines their productivity. Reproductive efficiency in a herd has been measured using different reproductive characteristics in cows, which has resulted in the development of different methods and standards for assessing the reproductive status of cattle. These methods include the measurement of simple parameters such as the interval between calves, open days, and the number of calves per year [34,35], the management of diseases, such as brucellosis and tuberculosis, and the recording of good livestock practices and consumption of concentrate with respect to the quantity of produced milk [34,49]. The reproductive efficiency of a herd was evaluated with the adoption and proper use of the reproductive records kept on each farm, since this is the only way to achieve a real table of the fertility of the herd [29,34,35,49].

### 2.3. Determination of Criteria and Sub-Criteria Weights

Following the AHP method, the weights for each criterion and sub-criterion were determined by a matrix structure with pairwise comparisons and checking that the overall comparisons were consistent [11,48].

These comparisons were made by a group of experts in the field (three zoo technicians, one production engineer, one administrative engineer, two forest engineers, and two environmentalists). To perform the weighting with AHP, they started by comparing, in pairwise matrices, the four main criteria. A similar process was then followed for the sub-criteria in each criterion.

Table 2 shows the AHP matrix with pairwise comparisons that the group of experts developed in order to obtain criteria weights. The result has a CR consistency indicator of 0.004 < 0.10. Note that the resulting weights for each criterion are the average weights of the normalized matrix. Similar matrices were applied to determine the weights for all sub-criteria but are not displayed in this paper in order to save space. The resulting weights for all criteria and sub-criteria are summarized in Table 5.

**Table 2.** Analytic hierarchy process (AHP) matrix for criteria weights (1 = equal importance, 3 = weak importance of one criterion with respect to another).

|  | **Financial** | **Technical** | **Social** | **Environment** |  |
| --- | --- | --- | --- | --- | --- |
| Financial | 1 | 2 | 3 | 3 |  |
| Technical | 0.5 | 1 | 2 | 2 |  |
| Social | 0.33 | 0.5 | 1 | 1 |  |
| Environment | 0.33 | 0.5 | 1 | 1 |  |
| Total | **2.16** | **4.0** | **7.0** | **7.0** |  |
|  | Normalized Matrix | | | | Average Weights |
| Financial | 0.46 | 0.50 | 0.43 | 0.43 | 0.4547 |
| Technical | 0.23 | 0.25 | 0.29 | 0.29 | 0.2630 |
| Social | 0.15 | 0.13 | 0.14 | 0.14 | 0.1411 |
| Environment | 0.15 | 0.13 | 0.14 | 0.14 | 0.1411 |
| Total | **1** | **1** | **1** | **1** |  |

### 2.4. Normalization of Criteria and Sub-Criteria Values

Data on criteria and sub-criteria were collected. Then, in order to adjust the values measured on different scales against a common scale, the obtained values were normalized using the maximum and

minimum values from all the analyzed alternatives. Then, the normalized values were put into the range [0,1] using the following formulae:

$$X = \frac{X - X_{\min}}{X_{\min} - X_{\max}} \text{ for values in which higher is better} \tag{1}$$

$$X' = \frac{X_{\max} - X}{X_{\max} - X_{\min}} \text{ for values in which higher is better} \tag{2}$$

where $X'$ is the normalized value, $X$ is the criterion value to be normalized, $X_{\max}$ is the maximum value within the criterion's data range, and $X_{\min}$ is the minimum value within the criterion's data range.

### 2.5. Calculation of the Integrated Sustainability Index

Table 3 shows how the sustainability indices corresponding to each of the criteria and the integrated sustainability index IISLE were determined.

**Table 3.** Calculation of sustainability indices corresponding to the economic, financial, technical, social, and environmental criteria.

| Calculation of Indicators | Conventions |
|---|---|
| $ISDF = \sum_{i=1}^{n} \left( ScF_{i(normalizado)} * P_{ij} \right)$ | **ISDF**: Sustainability indicator financial dimension.<br>**ScF$_i$**: Each of the proposed and standardized financial criteria.<br>**P$_{ij}$**: Weight of each ScFi (according to the AHP method).<br>**n**: Number of sub-criteria in the financial criteria. |
| $ISDT = \sum_{i=1}^{n} \left( ScT_{i(normalized)} * P_{ij} \right)$ | **ISDT**: Sustainability indicator technical dimension.<br>**ScT$_i$**: Each of the proposed and standardized technical criteria.<br>**P$_{ij}$**: Weight of each ScTi (according to the AHP method).<br>**n**: Number of sub-criteria in the technical criteria. |
| $ISDS = \sum_{i=1}^{n} \left( ScS_{i(normalized)} * P_{ij} \right)$ | **ISDS**: Sustainability indicator social dimension.<br>**ScS$_i$**: Each of the proposed and standardized social criteria.<br>**P$_{ij}$**: Weight of each ScSi (according to the AHP method).<br>**n**: Number of sub-criteria in the social criteria. |
| $ISDA = \sum_{i=1}^{n} \left( ScA_{i(normalized)} * P_{ij} \right)$ | **ISDA**: Sustainability indicator environmental dimension.<br>**ScA$_i$**: Each of the proposed and standardized environmental criteria.<br>**P$_{ij}$**: Weight of each ScAi (according to the AHP method).<br>**n**: Number of sub-criteria in the environmental criteria. |
| $IISLE = \sum_{i=1}^{m} \left( ISDi * Pij \right)$ | **IISLE**: Integrated indicator of sustainability in specialized dairy.<br>**ISD$_i$**: Sustainability indicators determined in each dimension.<br>**P$_{ij}$**: Weight of each ISDi (according to the AHP method).<br>**m**: Each of the indicators of the four criteria. |

## 3. Application: A Colombian Case Study

This section presents the validation of the proposed index. It includes a description of the farms selected and the information-gathering process.

### 3.1. Location and Description of Alternatives

The research was carried out at 18 farms located on the tropical plateau of the Antioquia state (although in Colombia these political divisions are called "departments"), 1800 m above sea level. The farms are dedicated exclusively to the production of milk and raising replacement heifers. They are located in the municipalities of "San Pedro de los Milagros" and "La Unión" (specifically, in Charco Verde, La Pulgarina, La Lana, El Tambo, Quebrada Negra, Las Acacias, La Concha, Buenavista, and La Almería). All farms have a system of intensive supplemental grazing. Sixty percent of the farms are at a high technological level and possess mechanical milking equipment, a milking parlor with adequate infrastructure, and a cold tank for the storage of milk for a period of time without loss due to

contamination or decomposition. Roads are built with cement to facilitate the comfortable movement from the paddocks to the milking site of purebred animals specialized in milk production. Management activities in terms of feeding are complex, with the use of highly concentrated supplements that meet the nutritional requirements of the cows that are in production.

The remaining 40% of farms are at a medium technological level. Some have mechanical but obsolete milking equipment, while others still perform milking manually. These farms also have a storage tank and adequate but somewhat rudimentary infrastructure. However, they use purebred animals specialized in milk production, and the management and maintenance of these animals are adequate.

The farms were selected by following a convenience sampling procedure since the authors contacted producers that were interested in participating in this study and had appropriate data. The farms can be classified according to the variables shown in Table 4.

**Table 4.** A description of each of the alternatives for selected farms.

| Alternative Number | Description |
| --- | --- |
| A1 | Small Holstein CBPG |
| A2 | Medium Holstein CBPG |
| A3 | Large Holstein CBPG |
| A4 | Small Holstein SBPG |
| A5 | Medium Holstein SBPG |
| A6 | Large Holstein SBPG |
| A7 | Small Jersey CBPG |
| A8 | Medium Jersey CBPG |
| A9 | Large Jersey CBPG |
| A10 | Small Jersey SBPG |
| A11 | Medium Jersey SBPG |
| A12 | Large Jersey SBPG |
| A13 | Small Holstein–Jersey CBPG |
| A14 | Medium Holstein–Jersey CBPG |
| A15 | Large Holstein–Jersey CBPG |
| A16 | Small Holstein–Jersey SBPG |
| A17 | Medium Holstein–Jersey SBPG |
| A18 | Large Holstein–Jersey SBPG |

- Herd size: According to the numbers of cows in production, either small (less than 26), medium (between 26 and 50), or large (more than 50) [50].
- Predominant breed: Only Holstein, only Jersey, or both Holstein and Jersey.
- Good livestock practices certificate (BPG): With a CBPG certificate or without a SBPG certificate.

*3.2. Data on Criteria and Sub-Criteria*

The information obtained in the semi-structured interview of each of the producers was compiled. The information related to the cost-generating elements was analyzed according to the absorption costing methodology proposed by [51], where the following were determined: Total production cost, cost per liter of milk, break-even point, and margin of security. Similarly, to determine financial criteria (gross margin, profitability, and B/C ratio), we used the methodology for financial indicators proposed by [52].

The information related to the social criteria was analyzed, taking into account the perception scale established for each of the selected criteria. A lack of generational relay was evident. In small- and medium-size producers, it is necessary to encourage generational relay in order to take advantage of older producers' experience and knowledge and combine them with younger producers' strength and new ideas [30,38]. Associations among producers allow them to improve relations and create a social consensus. According to the perceptions of producers, market access is threatened due to high

product transportation costs and low, uncompetitive prices. For the production and marketing of milk, they have institutional support from various governmental institutions in the country. On the other hand, producers perceive credit with financial entities to be difficult to access [5,32].

Likewise, regarding the information collected on the environmental criteria, the input for the "occupation" and "rest period" sub-criteria was the averaged data from each farm. Likert scales were used for the "protection and conservation of water sources" and "erosion" sub-criteria, while ordinal scales were used for the remaining criteria (load capacity, pesticide use, soil conservation practices, excreta treatment, wastewater destination, and density of worms [19]). For the technical criteria, data from the technical records of each farm were used for the related sub-criteria (liters cow/day, interval between births, open days, farm/year deliveries, tuberculosis and brucellosis certificates, good livestock practices, and milk/concentrate ratio).

### 3.3. Integrated Sustainability Index

For each of the financial, technical, social, and environmental criteria, a sustainability index was calculated, taking into account the non-correlated sub-criteria, which were expressed as a linear combination of the original sub-criteria. Once the sustainability index was obtained, an integrated sustainability index value was determined for each farm under analysis. The calculation of each of the indices (Table 3) and the integrated sustainability indicator is illustrated in the following section.

### 3.4. Statistical Analysis

A multiple correlation analysis and a cluster analysis were performed as part of the adopted methodology.

First, a multiple correlation analysis was carried out on the sub-criteria analyzed within each criterion in order to define the greatest possible variability in order that the least amount of information would be lost. From this analysis, it was possible to reduce the number of sub-criteria from 43 to 30. Subsequently, a multiple correlation analysis was performed between each of the sustainability indicators and the integrated indicator IISLE in order to determine which of the criteria explained, or had the most direct relationship with, the result of the index. We used the following formula:

$$\gamma = \frac{Sxy}{SxSy} \tag{3}$$

where $\gamma_{xy}$ is the correlation coefficient, $Sxy$ is the covariance of $xy$, and $SxSy$ is the product of the standard deviations of $x$ and $y$.

In order to determine the maximum degree of homogeneity between the farm groups (alternatives) with respect to the integrated sustainability index in specialized dairy (IISLE), a cluster or conglomerate analysis was carried out. This analysis allowed us to determine the possible groupings according to similarities and/or differences. This analysis was performed using the R language [53] and Euclidean distance:

$$d\,(Xi, Xj) = \sqrt{\sum_{c=1}^{p}(Xic - Xjc)^2}; p = subcriteria \tag{4}$$

where $d$ is the Euclidean distance and $Xi$, $Xj$ are sub-criteria.

## 4. Results and Discussion

The first result of the methodology's application was the determination of the weights for each criterion following the process described in Section 2.3. The resulting weights are shown in Table 5. Note that the sum of all weights in Level 1 is 100%, while in Level 2 the sum of the weights in each set of sub-criteria is 100%.

**Table 5.** Criteria and sub-criteria weights.

| Level 1 | | Level 2 | |
|---|---|---|---|
| **Criteria** | **Criteria Weight** | **Sub-Criteria** | **Sub-Criteria Weight** |
| Financial | 0.4547 | ScF1 | 0.205 |
| | | ScF2 | 0.148 |
| | | ScF3 | 0.120 |
| | | ScF4 | 0.120 |
| | | ScF5 | 0.174 |
| | | ScF6 | 0.121 |
| | | ScF7 | 0.111 |
| Technical | 0.2630 | ScT1 | 0.182 |
| | | ScT2 | 0.224 |
| | | ScT3 | 0.224 |
| | | ScT4 | 0.075 |
| | | ScT5 | 0.117 |
| | | ScT6 | 0.102 |
| | | ScT7 | 0.076 |
| Social | 0.1411 | ScS1 | 0.457 |
| | | ScS2 | 0.196 |
| | | ScS3 | 0.118 |
| | | ScS4 | 0.118 |
| | | ScS5 | 0.055 |
| | | ScS6 | 0.055 |
| Environmental | 0.1411 | ScA1 | 0.252 |
| | | ScA2 | 0.135 |
| | | ScA3 | 0.103 |
| | | ScA4 | 0.097 |
| | | ScA5 | 0.119 |
| | | ScA6 | 0.066 |
| | | ScA7 | 0.082 |
| | | ScA8 | 0.043 |
| | | ScA9 | 0.038 |
| | | ScA10 | 0.027 |
| | | ScA11 | 0.038 |

As seen in Table 5, the most relevant criterion is the financial one (45.5%), followed by the technical criterion (26.3%) and the environmental and social criteria, which have similar values (14.1% for each one). Studies carried out by [10] show slightly similar values (economic, 40%; technical, 30%; environmental, 20%; and social, 10%). However, these values differ greatly from the study conducted in Costa Rica by [11], where the weighting was: Technical criterion, 36%; economic criterion, 25%; ecological criterion, 24%; and social criterion, 15%. Likewise, the authors of [4] established the following weights: Social, 37%; economic, 33%; and environmental, 30%. The authors of [19] established the following weights: Technical criterion, 29%; economic and environmental criteria, 24%; and social criterion, 23%. It is important to mention, as observed in different studies, that a high degree of subjectivity is present when prioritizing the different analyzed criteria.

Table 6 summarizes the sustainability indices of the financial, technical, social, and environmental criteria, as well as the integrated sustainability index in specialized dairy (IISLE), in each of the farms (alternatives) analyzed in the study.

**Table 6.** Sustainability indices in specialized dairy production systems (SPLEs).

| FARM | ISDF | ISDT | ISDS | ISDA | IISLE |
|------|------|------|------|------|-------|
| A1 | 0.16349 | 0.45384 | 0.75150 | 0.64762 | 0.39448 |
| A2 | 0.76713 | 0.88460 | 0.73198 | 0.37431 | 0.72913 |
| A3 | 0.13502 | 0.57010 | 0.38071 | 0.58938 | 0.35138 |
| A4 | 0.55707 | 0.49826 | 0.09482 | 0.39051 | 0.44378 |
| A5 | 0.43295 | 0.55563 | 0.29942 | 0.50646 | 0.45810 |
| A6 | 0.44934 | 0.59458 | 0.35126 | 0.52682 | 0.49677 |
| A7 | 0.68636 | 0.71836 | 0.76308 | 0.43927 | 0.67015 |
| A8 | 0.53851 | 0.73492 | 0.62921 | 0.47152 | 0.58332 |
| A9 | 0.66183 | 0.57295 | 0.51916 | 0.68074 | 0.61246 |
| A10 | 0.83206 | 0.49715 | 0.48348 | 0.32036 | 0.61154 |
| A11 | 0.83495 | 0.53985 | 0.58166 | 0.49185 | 0.66383 |
| A12 | 0.86325 | 0.47041 | 0.58988 | 0.63020 | 0.68919 |
| A13 | 0.41815 | 0.54417 | 0.58988 | 0.63543 | 0.50475 |
| A14 | 0.39904 | 0.58814 | 0.83810 | 0.51838 | 0.53534 |
| A15 | 0.45441 | 0.77293 | 0.32980 | 0.75901 | 0.56269 |
| A16 | 0.72854 | 0.40350 | 0.39323 | 0.68446 | 0.57867 |
| A17 | 0.66158 | 0.50363 | 0.51652 | 0.44259 | 0.57547 |
| A18 | 0.64389 | 0.42896 | 0.43786 | 0.67103 | 0.57074 |

In Table 6, it can be seen that the highest IISLE value corresponds to the alternative *A2* (Medium Holstein CBPG). This farm also has the highest ISDT value, which was due mainly to the highest production of liters/animal/day, an interval between births of 7% above the optimum level, an optimal value on open days, and a milk/concentrate ratio equal to 4.0, and the fourth-best ISDF value, which was mainly due to the higher milk production per animal/year, a low production cost per liter of milk, a B/C ratio above 1, and an acceptable return value. It has a good ISDS value and the lowest value with respect to the ISDA. All of these characteristics make *A2* the most sustainable alternative.

The second position is occupied by the alternative *A12* (Large Jersey SBPG). This farm has the highest ISDF value, since it has the highest profit margin, the highest profitability, and a low production cost per liter of milk. However, this alternative had a poor result in the technical criterion. Additionally, it can be concluded from Table 6 that the least sustainable farm is *A3* (Large Holstein CBPG), since it has the lowest ISDF value, a high production cost per liter of milk, the lowest milk production per animal/year, a B/C ratio well below 1, and unsatisfactory technical parameters that prevent it from performing well.

It is important to note that the integrated indicator of sustainability in specialized dairy (IISLE) is very influenced by financial and technical criteria. According to the results shown in Table 6, characteristics such as farm size, type of breed, and a certificate of good livestock practices exert little influence on the IISLE. On a social level, it is necessary to mention that, at present, a lack of labor is one of the most serious problems in the sector. On the one hand, members of the working-class population are moving to the cities, leaving the countryside with a small workforce. On the other hand, young people do not see dairy farming as an attractive or profitable enough activity to work on.

In Figure 1, it can be seen that there is a growing linear relationship between ISDF and IISLE (the higher the financial sustainability index, the higher the integrated sustainability index). There is a similar relationship between ISDT and IISLE. However, the relationship between ISDS and ISDA with IISLE is not as linear, since the greater the dispersion between the points, the lower the intensity of the relationship between the IISLE response variable and the explanatory variables, therefore, the smaller the correlation. However, according to the result of the partial correlation, which for the four explanatory variables is 1, the four variables have the same level of importance to the integrated IISLE sustainability index.

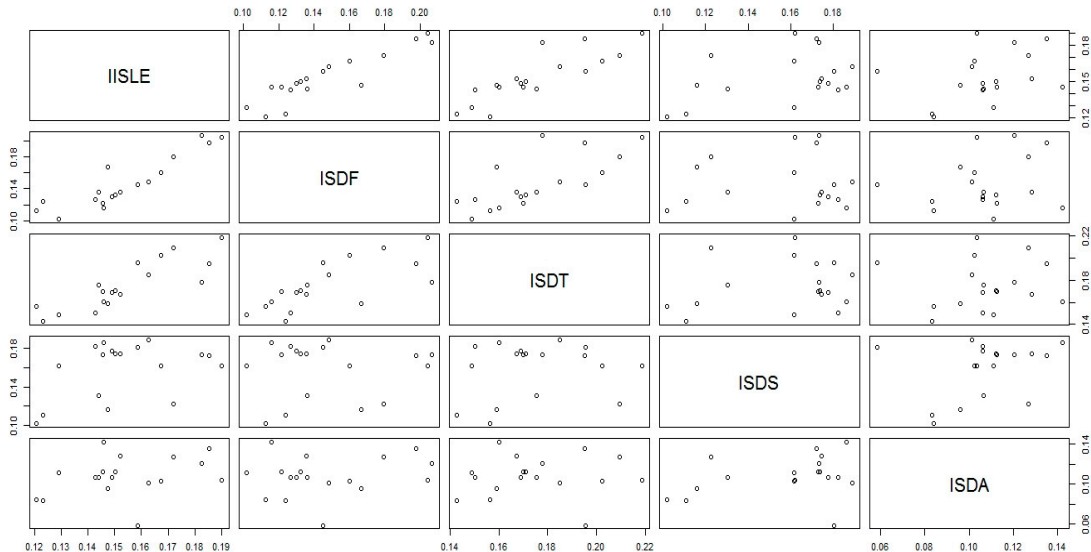

**Figure 1.** Correlation between the IISLE response variable and the four explanatory variables: Sustainability indicator financial dimension (ISDF), sustainability indicator technical dimension (ISDT), sustainability indicator social dimension (ISDS), and sustainability indicator environmental dimension (ISDA).

According to the cluster analysis, for the integrated sustainability indicator in specialized dairy (IISLE), it can be seen that four groups are generated in Figure 2. Cluster 1 groups together the alternatives *A8*, *A13*, *A14*, *A1*, and *A3*. Farm *A8* has very good performance at a technical level, which is represented in parameters such as good cow/year production, an acceptable interval between births, and open days, it is the fifth-best farm in terms of the ISDS, and in terms of the financial criteria it has an average indicator value due to its production cost and low profitability per animal. *A13* and *A14* have similar values of ISDF and ISDT, which are low compared with the average value of the farms. Farms *A1* and *A3* have the lowest integrated sustainability indicator values, which characterize them as being financially and technically unsustainable.

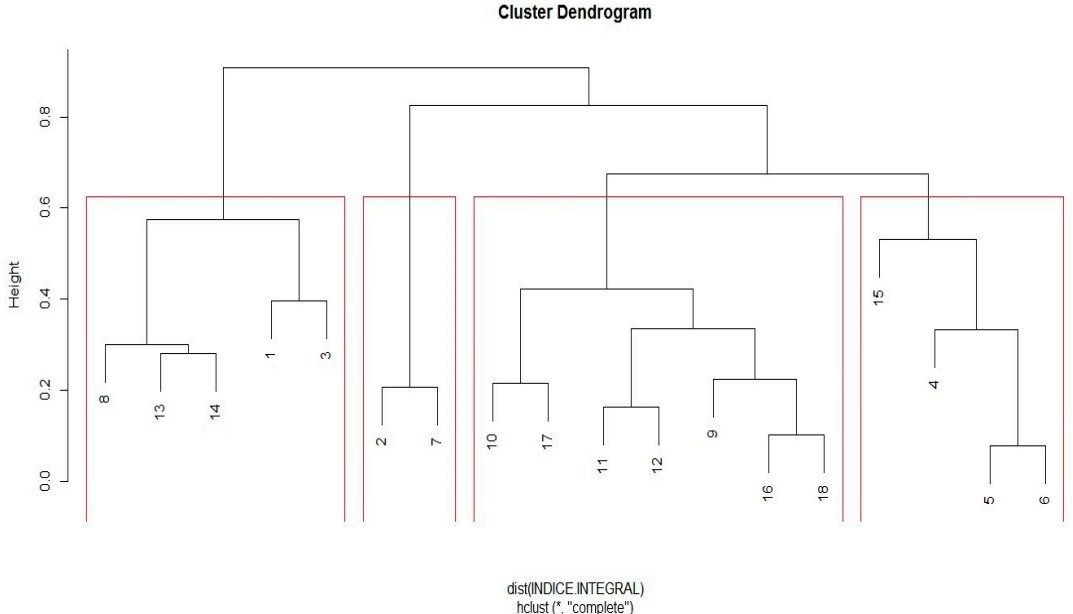

**Figure 2.** Cluster analysis for the integrated sustainability indicator in specialized dairy (IISLE).

The alternatives *A2* and *A7* are grouped into Cluster 2. These two farms are considered the most sustainable in terms of the four criteria. Farm *A2* is the most sustainable according to the result of IISLE. Farm *A7* occupies the third position and is surpassed by *A12* only in the financial criteria. A third cluster groups together the alternatives *A10*, *A17*, *A11*, *A12*, *A9*, *A16*, and *A18*. Farms *A10*, *A11*, and *A12* are the most financially sustainable, since they have high milk/cow/day production, a low production cost, and the best B/C ratio, which put them in positions 6, 4, and 2, respectively. Farm *A9*, which had a somewhat similar performance, occupies position 5 regarding integrated sustainability according to the IISLE integral sustainability indicator. Farms *A16*, *A17*, and *A18* show good financial performance and each has an IISLE value that is 3% above the average of all farms.

The fourth and final cluster is formed by the alternatives *A15*, *A4*, *A5*, and *A6*. These farms have values below the average of all farms in financial sustainability and satisfactory values regarding technical sustainability. Farms *A6*, *A5*, and *A4* occupy positions 14, 15, and 16, respectively, in the IISLE integral sustainability indicator.

## 5. Conclusions

- It is necessary to strengthen the sustainable development of specialized livestock. Therefore, these production systems must be analyzed and managed in an integrated manner and from the point of view of economic, technical, social, and environmental criteria. However, in order to achieve satisfactory changes in the sustainable development of specialized livestock in the high tropics, the actors involved (government, public and private institutions, consumers, suppliers, and livestock producers) must work together to support and strengthen not only the economic, technical, and social conditions of producers, as well as the conservation of the natural resources of the farms.

- It is essential to have a panel of expert evaluators determine the weights for criteria and sub-criteria, and to apply an appropriate prioritization method in order to reduce biases and subjectivity when determining the hierarchy of importance with respect to the criteria.

- When determining the sub-criteria within each criterion, it is important to find a balance in the number of sub-criteria in order to not generate a trend in the calculation of the integrated sustainability index and to be able to equitably cover the most relevant aspects at the financial, technical, social, and environmental levels.

- The integrated sustainability index in specialized dairy (IISLE) shows, in this study of 18 dairy farms located in Antioquia, Colombia, that production systems which have good financial and technical performance are more sustainable.

- Intensive livestock systems have raised problems about environmental impacts and food security for the past 20 years. As a consequence, there is a strong social demand for sustainable livestock systems, which should be environmentally friendly, economically viable for farmers, and socially acceptable, especially in terms of animal welfare.

**Author Contributions:** Conceptualization, G.P.R. and S.B.; Methodology, G.P.R. and S.B.; Data curation, G.P.R.; Formal analysis, G.P.R. and S.B.; Funding Acquisition, G.P.R.; Writing—original draft, G.P.R. and S.B.; Writing—review & editing, G.P.R. and S.B.; Supervision, S.B. All authors have read and agreed to the published version of the manuscript.

**Funding:** This research was funded by Universidad Nacional de Colombia, grant number Res. M. SFCA-004 of 2019.

**Acknowledgments:** We thank the livestock producers at the La García, Manantial de la Sierra, La Pulgarina, La Montañita, and La Esperanza farms in the municipality of San Pedro de los Milagros, the producers at the La Jacoba, La Virginia, La Pradera farms, and the Gómez Osorio family in the municipality of Unión Antioquia, Colombia for their time, dedication, and information.

**Conflicts of Interest:** The authors declare no conflict of interest.

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
