# Peer review of "An Integrated Indicator to Analyze Sustainability in Specialized Dairy Farms in Antioquia—Colombia"

_sustainability, doi:10.3390/su12229595_

Round 1

Reviewer 1 Report

The work is interesting to the readers of the journal. English writing is OK, but Editorial improvement is needed.

Title reads odd with a comma. Better to revise as “An Integrated Indicator to analyze Sustainability in Specialized Dairy Farms in Antioquia- Colombia.

L32. Check the citation style of Sustainability!

L89. Are all the citations needed? Better to cite fewer critical papers.

L94. Table numbering is not that way for journal papers. It reads confusing to present the weighing results in the table.

L121, 125, 129 and many other places. There are incorrect layouts around citations.

L245. Better to combine table 4.1 and 4.2.

LL333-336. While you are expecting environmentally friendly, economically viable for farmers and socially acceptable, is it better for you to briefly mention that incorporating best manure management practices would be helpful to reach the goal. Your environmental factor does not consider some manure issues, such as greenhouse emission, and unpleasant odor concern. Please cite Waldrip et al. (2020. Animal Manure: Production, Characteristics, Environmental Concerns and Management. ASA Special Publication 67. DOI:10.2134/asaspecpub67) to address it with a couple of sentences if not more.

Author Response

The paper has been language edited

Title reads odd with a comma. Better to revise as “An Integrated Indicator to analyze Sustainability in Specialized Dairy Farms in Antioquia- Colombia.

The title has been changed

L32. Check the citation style of Sustainability!

The citation style has been changed to the Sustainability style. Quote numbers are in order of appearance.

L89. Are all the citations needed? Better to cite fewer critical papers.

Citations were reviewed

L94. Table numbering is not that way for journal papers. It reads confusing to present the weighing results in the table.

Table numbering style has been changed.

Tables were organized

In order to reduce confusion, weighing results are shown in another table (table 5), explaining how weights of criteria and sub-criteria add up.

Reviewer 2 Report

Although it reports on work that is relevant to the field of Sustainability. The coverage and content of the manuscript are balanced and both methodology and results are clearly described.  Although the manuscript is generally readable, it needs language editing.

For example:

Line 228: it is: "results session and discussion". It should read: "results section and discussion"

Line 237: It is: "Table 4.1 236 y 4.2 shows", It should read: "Table 4.1 236 and 4.2 show"Figure 4.1 is not readable, it is too small to see the units and descriptions.

Author Response

Although it reports on work that is relevant to the field of Sustainability. The coverage and content of the manuscript are balanced and both methodology and results are clearly described.  Although the manuscript is generally readable, it needs language editing.

For example:

Line 228: it is: "results session and discussion". It should read: "results section and discussion"

The line has been corrected

Line 237: It is: "Table 4.1 236 y 4.2 shows", It should read: "Table 4.1 236 and 4.2 show"Figure 4.1 is not readable, it is too small to see the units and descriptions.

The lines have been corrected

Font size in table 4.1 (now table 1) was enlarged, and information on weights was taken away to be shown in a new table (table 5)

Figure 4.1 (now figure 1 ) has been improved and correlation analysis has been explained in the text

The document has been language edited.

Reviewer 3 Report

The manuscript “Integrated Indicator to analyze Sustainability in Specialized Dairy Farms, Application to Antioquia-Colombia.” was not adequately prepared. The authors have not put any attention to the presentation of their ideas. All references, figures, and tables were disorganized. The introduction and methods sections were poorly written. The results were not relevant since the authors did not correctly describe the statistical methods. Some comments Line 13: you might not need to state the Spanish name in the abstract Line 15-17: might re-text as “In the current study, the information from 18 dairy farms in a Colombian 16 milk production region were used to validate the IISLE. Might be named the region if it is not long Line 19 and 20: which software was used for analyses, the authors did not need to capitalize on the name of the method Line 19 which type of correlation Line 20-24: it is hard to link the results with the conclusion Line 32 Something wrong with the references Line 32 “This means that the approach that is” please re-text Line 40: the sentence does not make sense; you might list the criteria here Line 41: You do not need to define SDPS again Line 45: I do not think it is the right scientific way to write Line 59-66: should be in the introduction Line 67-80: I am not sure the authors describe the methods or reviewing the literature; this section should be in the introduction as well Line 82: the citation is not suitable; the sentence wrote carelessly Table 2.1 How did the authors define the weight Where is table 1 What does 45,4% come from? Using 45.4 instead of 45,4 The authors should begin with the farm selections as the authors have different types of farms; the information about the farm is not clear. Section 3.4: please give more information about statistical analyses. Where are the figure 1,2 and 3 in the manuscript?

Author Response

The paper has been reviewed and is better organized.

The abstract has been reviewed

The paper has been language edited

References, tables and figures were organized

The statistical methods have been described

The software used in the analysis has been specified (R software)

The Farm selection section has been improved

Round 2

Reviewer 3 Report

All my comments have been addressed. The manuscript has improved. However, extensive editing of the English language and style required. The figures need to have a better resolution and please reformat the table. 

Author Response

We improved layout

All tables have been reformatted and reviewed. Figures were changed. Here are some specific details:

Table 1. Merged some cells in the third column and vertically centered texts for all cells in the second and third column

Table 2. Centered some texts and retired background color that was in the left side of the first matrix.

Table 3. Cells in the first column are equations from the word equation editor, by default the equation editor writes in Italic letters and we do not know how to change them. However in the second column we changed all words that were in italic to normal. We centered equations.

Table 4. We changed text in the first column from italic to normal.

Table 5. We centered texts

Table 6. We changed text in the first column from italic to normal, and centered texts.

Figure 1. We changed to an image with better resolution

Figure 2. We changed to an image with better resolution

English was reviewed. The manuscript has been reviewed by the MDPI English Editing Service. (English editing ID: English-23745).
